# The Effect of Electroencephalography Abnormalities on Cerebral Autoregulation in Sedated Ventilated Children

Madhuradhar Chegondi [1,2,*], Wei-Chiang Lin [3], Sayed Naqvi [4], Prithvi Sendi [5,6]
and Balagangadhar R. Totapally [5,6]

1 Division of Critical Care Medicine, Stead Family Children's Hospital, Iowa City, IA 52242, USA
2 Department of Pediatrics, Carver College of Medicine, University of Iowa, Iowa City, IA 52242, USA
3 Department of Biomedical Engineering, Florida International University, Miami, FL 33174, USA
4 Department of Neurology, Nicklaus Children's Hospital, Miami, FL 33155, USA
5 Division of Critical Care Medicine, Nicklaus Children's Hospital, Miami, FL 33155, USA
6 Herbert Wertheim College of Medicine, Florida International University, Miami, FL 33199, USA
* Correspondence: madhuradhar-chegondi@uiowa.edu; Tex.: +1-319-356-1615; Fax: +1-319-356-8443

**Abstract:** Purpose: To determine the effects of non-ictal electroencephalogram (EEG) changes on cerebrovascular autoregulation (AR) using the cerebral oximetry index (COx). Materials and Methods: Mean arterial blood pressure (MAP), cerebral tissue oxygenation (CrSO2), and EEG were acquired for 96 h. From all of the EEG recordings, 30 min recording segments were extracted using the endotracheal suction events as the guide. EEG recordings were classified as EEG normal and EEG abnormal groups. Each 30 min segment was further divided into six 5 min epochs. Continuous recordings of MAP and CrSO2 by near-infrared spectroscopy (NIRS) were extracted. The COx value was defined as the concordance (R) value of the Pearson correlation between MAP and CrSO2 in a 5 min epoch. Then, an Independent-Samples Mann-Whitney U test was used to analyze the number of epochs within the 30 min segments above various R cutoff values (0.2, 0.3, and 0.4) in normal and abnormal EEG groups. A *p*-value < 0.05 was considered significant, and all analyses were two-tailed. Results: Among 16 sedated, mechanically ventilated children, 382 EEG recordings of 30 min segments were analyzed. The proportions of epochs in each 30 min segment above the R cutoff values were similar between the EEG normal and EEG abnormal groups (*p* > 0.05). The median concordance values for CSrO$_2$ and MAP in EEG normal and EEG abnormal groups were similar (0.26 (0.17–0.35) and 0.18 (0.12–0.31); *p* = 0.09). Conclusions: Abnormal EEG patterns without ictal changes do not affect cerebrovascular autoregulation in sedated and mechanically ventilated children.

**Keywords:** cerebrovascular autoregulation; cerebral oximetry index; EEG changes; sedatives; critically ill; children

## 1. Introduction

Impaired cerebral autoregulation (AR) in children with traumatic brain injury (TBI) is associated with poor outcomes [1,2]. Cerebral AR refers to the physiological ability of the body to maintain steady cerebral perfusion in response to changes in cerebral perfusion pressure (CPP) by modulating vascular resistance through arteriolar caliber [3]. Multiple mechanisms in the cerebral vasculature regulate cerebral blood flow (CBF), including pressure and metabolic autoregulation, reactivity to hypoxia, hypercarbia, and hypoglycemia [4].

Measuring and monitoring AR can be done clinically at the bedside by measuring changes in CBF and cerebral blood volume (CBV), or with CBF surrogates to CPP [3,5]. Static and dynamic methods are utilized for measuring AR. The static assessment relates to the changes in CBF at a steady state when the arterial blood pressure (ABP) varies [4,5]. Dynamic assessment involves the continuous monitoring of CBF changes by assessing the amplitude variability of ABP oscillations [6].

Cerebral oximetry based on near-infrared spectroscopy (NIRS) is a non-invasive method to measure regional cerebral tissue oxygen saturation (CrSO2) [7]. The cerebral oximetry index (COx) is an autoregulatory vasoreactivity derived from the correlation between mean arterial blood pressure (MAP) and CrSO2 in the time-domain analysis [8]. A COx value of zero suggests no correlation between MAP and CrSO2; it indicates an intact AR, and a value close to 1 suggests a strong correlation, indicating an impaired AR or MAP value beyond the autoregulation range [9]. COx cutoff values between 0.2 and 0.4 have been used as the threshold levels for detecting impaired cerebral autoregulation [8,10–13]. Continuous electroencephalogram (EEG) monitoring has been used in critically ill children who are at risk of neurological adverse events [14]. To evaluate the neurovascular coupling alterations, multimodal neuromonitoring using EEG and NIRS was previously used in children with seizures [15]. However, the association of EEG abnormalities with AR in critically ill children without ictal changes is unknown. The objective of our study was to determine the effect of abnormal EEG patterns on AR in sedated, mechanically ventilated children and determine the effect on AR at various concordance levels. We hypothesized that non-ictal EEG abnormalities are associated with impaired cerebral AR.

## 2. Materials and Methods

### 2.1. Design

A prospective observational multimodal neuromonitoring study.

### 2.2. Setting

Pediatric intensive care unit (PICU) at a tertiary care children's hospital.

### 2.3. Participants

Children aged 0–18 years were sedated, mechanically ventilated, on continuous EEG monitoring, and had an arterial line inserted for monitoring.

### 2.4. Procedures

We performed an analysis of data from a prospective observational multimodal neuromonitoring study data, funded by the US Department of Defense (Award# W81XWH-09-1-0295) under the original institutional review board (IRB) proposal. The original study's (Multimodal Neuromonitoring in Critically Ill Children) objective was to determine the prevalence of adverse neurological events (seizures, periodic epileptiform discharges, etc.) in critically ill children, and to identify associations between various physiological variables and those events [16]. Children with underlying cardiac conditions, children on high-frequency oscillatory ventilation, and those evaluated for brain death were excluded from the study. The study was approved by the IRB, and consent from a parent or a legal guardian was obtained. We reported the study details in a previous publication [16]. During the study period, the children were on continuous infusions of one or more sedatives such as midazolam, dexmedetomidine, ketamine, and/or pentobarbital.

### 2.5. Data Acquisition and Processing

Electrocardiogram, arterial blood pressure, and pulse oximetry waveform data were recorded using a standard bedside monitoring system (MP70, Philips Health Care Inc., Andover, MA, USA). These data were initially stored at a central station (Philips Health Care Inc., Andover, MA, USA) and transferred to a network drive every 24 h during the study period. In addition, numerical physiological data such as heart rate were exported continuously from the same MP70 in standard HL7 format at a frequency of 0.2 Hz. These numerical data were stored on the hospital network, and then exported manually to the data processing workstation. Regional oxygenation data ($CSrO_2$) were recorded from the cerebral sensor, placed on the frontal area using an INVOS Oximeter (Somanetics Corporation, Troy, MI, USA), and saved as 0.2 Hz continuous data streams. Finally, ventilator data were acquired from the ventilator's RS-232 interface (Servo-i, MAQUET Medical

Systems, Wayne, NJ, USA). Data from the INVOS oximeter and ventilator were recorded and stored on a portable personal computer and then transported to the data processing workstation. Simultaneously, continuous EEG monitoring data was recorded for the entire study duration. All recording devices were time-synchronized with the hospital's electronic clock [16]. Endotracheal (ET) suction events were used as a guide to extract the data needed for this study. The timing of all ET suctioning events was identified by reviewing the video of EEG monitoring. Thirty minutes of MAP and $CSrO_2$ were extracted before and after the time point of each ET suctioning event. These data were averaged to 1 value per 10 s and resampled at 0.1 Hz to reduce high-frequency noise and facilitate temporal correlation analysis [17]. The COx was calculated from each epoch recording of MAP and CrSO2 using the Pearson correlation analysis, which represented the AR status of the study subject during the recording period.

### 2.6. EEG Analysis

One of the authors (a neurologist) reviewed the EEG recordings (Natus Medical Inc., Middleton, WI, USA) during each 30 min recording and reported the EEG findings. The reviewing neurologist was blinded to the clinical data and other physiologic changes during the recording period. To understand the impact of EEG characteristics on AR status, each 30 min epoch was grouped into EEG normal and EEG abnormal. There were no electrical or clinical seizures during any of the 30 min segments analyzed in the study.

### 2.7. Statistical Analysis

Descriptive data are presented as medians with an interquartile range for continuous data and as frequencies for categorical data. In addition, a comparative analysis was performed between EEG normal and EEG abnormal groups.

Each 30 min recording was divided into six 5 min epochs, and $COx_5$ was calculated for each 5 min epoch. To identify the AR-impaired epochs, we have used the cutoff concordance threshold (R cutoff) values of 0.2 [8,10–13,15–18]. The 5 min epochs with $COx_5$ greater than the R cutoff were considered AR-impaired, and those less than or equal to the R cutoff as AR-intact. Independent-Samples Mann-Whitney U test was used to analyze the number of epochs within each 30 min segment above the R cutoff in both EEG groups. Similarly, we have compared the number of AR-impaired 5 min epochs in each 30 min segment in both EEG groups with $COx_5$ concordance thresholds at 0.3 and 0.4. To compare the differences in the concordance values in each EEG group, the average concordance in each 30 min segment was calculated from the $COx_5$ values, and the median concordance value of 30 min segments in EEG normal and EEG abnormal groups were compared using the Mann-Whitney U test. A *p*-value < 0.05 was considered significant, and all analyses were two-tailed. The COx was calculated from each epoch recording of MAP and CrSO2 using the Pearson correlation analysis, which represented the AR status of the study subject during the recording period. The correlation coefficient derived from Pearson correlation was the concordance level.

### 3. Results

We collected data from a cohort of 16 sedated and mechanically ventilated children. Of them, 13 (82%) were males. The median age was 3 (2–8) years. Two patients were managed for status epilepticus (without active clinical or electrographic seizures during the study period), and the rest had no other neurological condition including TBI, as a reason for admission. All patients received midazolam infusion; 12 were on fentanyl, 2 were on pentobarbital, 2 were on dexmedetomidine, and 4 patients were on ketamine infusion. In addition, 6 patients were on a paralytic medication infusion.

Continuous recordings of MAP and CSrO2 by NIRS were extracted from 332 ET suction events. There were no episodes of hypotension during the study period. From our cohort, two 30 min periods of observation were conducted before and after each of these 332 ET suction events. This yielded 664 30 min periods of observation. However,

we excluded the observations if there were any missing or incomplete EEG data during the entire 30 min of the observation period. As a result, 580 EEG recordings (30 min each) were available for review. From these 580 EEG recordings, we have excluded missing or incomplete NIRS or ABP variables during the entire 30 min of the observation period. Thus, a total of 382 EEG recordings with complete and usable data were included in the COx analysis. Among these recordings, 357 (93.5%) were in EEG normal group, and the remaining 25 (6.5%) were in EEG abnormal group (Table 1).

**Table 1.** Classification of electroencephalogram recordings with various rhythms.

| EEG Group | EEG Rhythm | N (%) |
|---|---|---|
| EEG-normal | Normal physiological | 244 (63.9) |
| | Prolonged suppression due to sedative effect | 113 (29.6) |
| EEG-abnormal | | 8 (2.1) |
| | Burst suppression | 12 (3.1) |
| | Periodic lateral epileptiform discharges | 5 (1.3) |
| | Increased activity other than seizure | |
| Total | | 382 (100%) |

EEG-electroencephalogram.

The median number of 5 min epochs above the R cutoff value of 0.2 was similar in both groups (Table 2; $p = 0.097$).

**Table 2.** The median (interquartile range) AR-impaired 5-min epochs between EEG normal and EEG abnormal groups at various COx concordance threshold values.

| Concordance Threshold Value | EEG Normal | EEG Abnormal | *p*-Value |
|---|---|---|---|
| 0.2 | 2 (2–3) | 2 (1–3) | 0.097 |
| 0.3 | 2 (1–3) | 2 (1–2.5) | 0.096 |
| 0.4 | 2 (1–2) | 1 (0.5–2) | 0.11 |

AR-autonomic regulation, EEG-electroencephalogram, COx-cerebral oximetry index.

At a COx concordance threshold of 0.3 or 0.4, the number of AR-impaired epochs was also similar in the two groups (Table 2). Similarly, the median of the average concordance value in the 30 min segments in our study was 0.25 (0.16–0.34). The medians of average concordance of 6 epochs in each 30 min segment were similar between the EEG normal and EEG abnormal groups [0.26 (0.17–0.35) vs. 0.18 (0.12–0.31); $p = 0.097$].

## 4. Discussion

In our prospective observational study, in critically ill children without acute brain injury, abnormal EEG patterns in the absence of seizures were not associated with impaired AR when tested at various thresholds of COx concordance levels. The number of epochs at various concordance thresholds was similar in both groups, indicating that abnormal EEG patterns did not impair the AR. NIRS-based COx has been validated for evaluating AR in animal models and adult patients [17,18]. In addition, NIRS-based COx monitoring is found to be a reliable tool for estimating optimal CPP [17]. In previous studies, the R cutoff value of less than 0.3 was used to represent intact AR and was sensitive to detect impaired AR [8,10]. However, the R cutoff value for the detection of impaired AR needs to be better established and varied in many studies [11,12]. For example, a COx value of 0.4 was used in children [11] and adults [12] undergoing cardiopulmonary bypass. Similarly, a recent study among children on extracorporeal membrane oxygenation support reported that the COx thresholds of 0.2 or 0.4 had shown comparable cerebral AR [13]. When we analyzed the data at various concordance thresholds of 0.2, 0.3, or 0.4, the abnormal EEG patterns did not show impairment in AR.

EEG waveform patterns may be classified into ictal changes, epileptogenic, and non-epileptogenic abnormalities [19]. EEG waveform patterns with spike and wave patterns, PLEDS, and generalized periodic discharges are considered epileptogenic waveforms, whereas diffuse slowing and burst suppression are non-epileptiform abnormalities [19]. We have included EEGs that were normal or showed increased sleep spindles in EEG in the normal group and those with abnormal patterns, epileptogenic, or non-epileptogenic EEG waveforms in EEG abnormal group. There were no ictal changes observed in our study. The concurrent NIRS and vEEG multimodal monitoring approach were previously used to analyze the effect of seizures on AR [20]. Similarly, NIRS-based CBV changes at the seizure onset and with epileptogenic focus have been described in patients with various seizure types [21–23]. However, the effects of EEG abnormalities without clinical or electrographic seizures on AR have not been reported. We report in our study that non-ictal EEG abnormalities are not associated with impaired cerebral AR.

In general, sedation has not been shown to affect cerebral AR. In our study, sedation was used in all patients, and midazolam and pentobarbital were used to control seizures in two patients. In a study, Vavilala et al. reported no AR changes with sevoflurane in non-TBI children undergoing elective surgical procedures [24]. Another study reported preserved AR with the use of propofol and remifentanil [25]. Using transcranial doppler and pressure reactivity index (PRx), pentobarbital-associated EEG burst suppression has shown improved AR in severe TBI patients [26]. An increased sedative effect could be a possible explanation for lower concordance in abnormal EEG groups in our study.

Besides NIRS-based COx, other non-invasive method AR indices such as mean velocity index, hemoglobin volume index, PRx, invasive jugular venous oximetry, and brain tissue oxygenation were used to assess cerebral AR continuously [4,27]. However, all these indices have methodological assumptions and drawbacks [28]. NIRS-based monitoring is technically easy to apply, non-invasive, does not require patient immobility, and measures dynamic cerebrovascular reactivity [27]. Nonetheless, NIRS monitors placed on frontal areas may not reflect other brain regions' vasoreactivity [27]. Furthermore, although it is widely used, COx is validated mainly in animal studies, and it may be affected by cerebral oxygen consumption [17]. Therefore, a universal threshold for detecting loss of cerebral AR is not established. As this technology is readily available and non-invasive, further studies need to be performed to establish a threshold concordance level for detecting AR impairment in various clinical conditions.

Our study being prospective eliminates the data acquisition/extraction limitations associated with retrospective studies. In addition, we maintained MAP and PaCO2 levels within the normal range while assessing the COx. The limitations include a small sample size and our study included children with various clinical diagnoses.

## 5. Conclusions

Our study showed that abnormal EEG findings without clinical or electrographic seizures are not associated with impaired cerebral AR in sedated and mechanically ventilated children without acute brain injury. The COx values were low in critically ill sedated children without neurological injury, or those with status epilepticus during seizure-free intervals, indicating preserved cerebral autoregulatory function. Our study adds pertinent information to the growing literature on the utility of NIRS in neuromonitoring. The use of COx as a tool in critically ill children needs further studies before its clinical application.

**Author Contributions:** M.C.: Conceptualization, Data curation, Formal analysis, Investigation, Methodology, Validation, Visualization, Writing—original draft, Writing—review & editing, and final approval of the version to be published; W.-C.L.: Data curation, Formal analysis, Software, Validation, Visualization, Writing—review & editing, and final approval of the version to be published; S.N.: Formal analysis, Investigation, Methodology, Writing—review & editing, and final approval of the version to be published; P.S.: Data curation, Investigation, Writing—review & editing, and final approval of the version to be published; B.R.T.: Conceptualization, Data curation, Formal analysis, Funding acquisition, Investigation, Methodology, Project administration, Resources, Software, Super-vision, Validation, Visualization, Writing—review & editing, and final approval of the version to be published. All authors have read and agreed to the published version of the manuscript.

**Funding:** This study was funded by the United States Department of Defense (*Award#W81XWH-09-1-0295*).

**Institutional Review Board Statement:** The study was conducted according to the guidelines of the Declaration of Helsinki and approved by the Institutional Review Board of Nicklaus Children's Hospital (WIRB # 1105045).

**Informed Consent Statement:** Informed consent was obtained from the participant children's parents or legal guardians involved in the study.

**Data Availability Statement:** The datasets generated during and/or analyzed during the current study are available from the corresponding author upon reasonable request.

**Acknowledgments:** Preliminary results were presented as a poster at the Neuro Critical Care Society meeting in September 2018, FL, USA.

**Conflicts of Interest:** The authors declare no conflict of interest. The funders had no role in the design of the study; in the collection, analyses, or interpretation of data; in the writing of the manuscript, or in the decision to publish the results.

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
