# Peer review of "The Effect of Electroencephalography Abnormalities on Cerebral Autoregulation in Sedated Ventilated Children"

_pediatrrep, doi:10.3390/pediatric15010002_

Round 1
Reviewer 1 Report
The authors try to determine the effects of non-ictal electroencephalogram (EEG) changes on cerebrovascular autoregulation (AR) using the cerebral oximetry index (COx). And the conclusion is that abnormal EEG patterns without ictal changes do not affect cerebrovascular autoregulation in sedated and mechanically ventilated children. There are some questions about this article.
1. The authors should also compare the effect of AR in patients with ictal EEG.
2. The authors should clarify the effect of different sedative drugs.
3. Is there any results from the literature showing that non-ictal EEG was related to AR?
Author Response
Reviewer 1:
The authors try to determine the effects of non-ictal electroencephalogram (EEG) changes on cerebrovascular autoregulation (AR) using the cerebral oximetry index (COx). And the conclusion is that abnormal EEG patterns without ictal changes do not affect cerebrovascular autoregulation in sedated and mechanically ventilated children. There are some questions about this article.
Comment 1: The authors should also compare the effect of AR in patients with ictal EEG.
Response: Thank you for the comment.
Our cohort did not have any patient with ictal periods during monitoring time.
Comment 2: The authors should clarify the effect of different sedative drugs.
Response: Thank you for the thoughtful suggestion. Many patients were on multiple sedative medications simultaneously and data can not differentiate sedative effect of individual medication.
Comment 3: Is there any results from the literature showing that non-ictal EEG was related to AR?
Response: After extensive search, we couldn’t find any pertinent literature. However, AR impairment well reported in interictal slow phase in known epilepsy patients.
Reviewer 2 Report
This study recorded MAP and CrSO2 using near-infrared spectroscopy (NIRS) and measured EEG changes in sedated and mechanically ventilated children, and revealed that abnormal EEG patterns without ictal changes do not affect cerebrovascular autoregulation. It provides important information for the clinical application of NIRS in neuromonitoring and the use of COx as a non-invasive cerebrovascular autoregulation index.
I would like to recommend that the authors provide more details of the medical equipment used in the study, such as the manufacturer and the model. Such information would facilitate the comparison of results obtained by other researchers in the same field.
Author Response
Reviewer 2:
This study recorded MAP and CrSO2 using near-infrared spectroscopy (NIRS) and measured EEG changes in sedated and mechanically ventilated children, and revealed that abnormal EEG patterns without ictal changes do not affect cerebrovascular autoregulation. It provides important information for the clinical application of NIRS in neuromonitoring and the use of COx as a non-invasive cerebrovascular autoregulation index.
I would like to recommend that the authors provide more details of the medical equipment used in the study, such as the manufacturer and the model. Such information would facilitate the comparison of results obtained by other researchers in the same field.
Response: We are very thankful for the feedback.
We have added the following paragraph in the methods section under the data acquisition and processing.
“Electrocardiogram, arterial blood pressure, and pulse oximetry waveform data were recorded using a standard bedside monitoring system (MP70, Philips Health Care Inc., USA). These data were initially stored at a central station (Philips Health Care Inc., USA) and then transferred to a network drive every 24 hours during the study period. In addition, numerical physiological data such as heart rate were exported continuously from the same MP70 in standard HL7 format at a frequency of 0.2 Hz. These numerical data were stored on the hospital network and then exported manually to the data processing workstation. Regional oxygenation data (CSrO2) were recorded from cerebral sensor, placed on the frontal area, using an INVOS Oximeter (Somanetics Corporation, Troy, MI, USA), and saved as 0.2 Hz continuous data streams. Finally, ventilator data were acquired from the ventilator's RS-232 interface (Servo-i, MAQUET Medical Systems, USA). Data from the INVOS oximeter and ventilator were recorded and stored on a portable personal computer and then transported to the data processing workstation. Simultaneously, continuous EEG monitoring data was recorded for the entire study duration. All recording devices were time-synchronized with the hospital's electronic clock.”
Reviewer 3 Report
The manuscript is well written and the data are of interest. This manuscript can be already accepted for publication.
Author Response
Reviewer 3:
Comment: The manuscript is well written and the data are of interest. This manuscript can be already accepted for publication.
Response: Thank you so much for your feedback.
Reviewer 4 Report
The paper present results of application of EEG and NIRS techniques to assess effect of EEG abnormalities on cerebral autoregulation in sedated ventilated children. The topic of the study is very important and very ambitious. The paper is well prepared but following update should be done before the next submission:
1) please clarify "surrogates to CPP"
2) please clarify definitions: "The static assessment relates to arterial blood pressure (ABP) and CBF changes at a steady-state [4, 5]. Dynamic assessment involves continuous monitoring of AR by assessing the rate of change with the measurement of ABP and CBF [6]."
3) please clarify statements “The objective of our study was to determine the effect of abnormal EEG patterns on the AR in sedated mechanically ventilated children and determine the effect on AR at various concordance levels. We hypothesized that the non-ictal EEG abnormalities are associated with impaired cerebral AR.”
4) please describe technical details of “A prospective observational multimodal neuromonitoring study”
5) please clarify “Endotracheal (ET) suction events were used as the guide to extract the data needed for this study”
6) please explain the reason of 30 minutes testing referring to “Thirty minutes of MAP and CSrO2 were extracted before and after the time point of each ET suctioning event."
7) please explain what does mean “temporal correlation analysis”
8) please give more technical details referring to "The COX was calculated from each epoch recording of MAP and CrSO2 using the Pearson correlation analysis"
9) please give technical data related to EEG device and NIRS device. Also, please clarify the problem of synchronization between EEG and NIRS devices.
10) please clarify "EEG normal and EEG abnormal" (the criteria for classification)
11) please explain the reason of application of "Mann-Whitney U test or Median test"
12) please clarify description given in lines 112-115
13) please clarify relation between “16 sedated and ventilated children” and "Continuous recordings of MAP and CSrO2 by NIRS were extracted from 332 ET suction events" and "From these 332 ET suction events, a total of 580 EEG recordings (30 minutes each) were available for review" and “for 96 hours" (line 15)
14) please clarify description given in line 139
15) please update Introduction by declaring whether method proposed in this study is novel one
16) please explain “PaCO2” (line 195)
17) please specify Conclusions
18) please clarify descriptions given in line 24-25
19) please update description of References
Author Response
Reviewer 4:
The paper present results of application of EEG and NIRS techniques to assess effect of EEG abnormalities on cerebral autoregulation in sedated ventilated children. The topic of the study is very important and very ambitious. The paper is well prepared but following update should be done before the next submission:
Comment 1: 1) please clarify "surrogates to CPP"
Response: We apologize for not being clear.
We edited to the following sentence-
“Measuring and monitoring AR can be done clinically at the bedside by measuring changes in CBF and cerebral blood volume (CBV) or with the CBF surrogates to CPP.”
Comment 2: 2) please clarify definitions: "The static assessment relates to arterial blood pressure (ABP) and CBF changes at a steady-state [4, 5]. Dynamic assessment involves continuous monitoring of AR by assessing the rate of change with the measurement of ABP and CBF [6]."
Response: We edited to the following-
The static assessment relates to the changes in CBF at a steady-state when arterial blood pressure (ABP) varies.
Dynamic assessment involves continuous monitoring of CBF changes by assessing the amplitude variability of the ABP oscillations.
Comment 3: 3) please clarify statements “The objective of our study was to determine the effect of abnormal EEG patterns on the AR in sedated mechanically ventilated children and determine the effect on AR at various concordance levels. We hypothesized that the non-ictal EEG abnormalities are associated with impaired cerebral AR.”
Response: We classified EEG patterns other than normal and sedation related EEG suppression patterns as abnormal EEG patterns, which includes burst suppression, periodic lateral epileptiform discharges (PLEDs), and increased activity other than seizures. The objective of our study is to evaluate these abnormal EEG patterns effect on the cerebral AR at various concordance ( R) cutoff values (0.2, 0.3, and 0.4) when compared to the normal patterns.
Comment 4: 4) please describe technical details of “A prospective observational multimodal neuromonitoring study”
Response: We have added the following paragraph in the methods section under the data acquisition and processing.
“Electrocardiogram, arterial blood pressure, and pulse oximetry waveform data were recorded using a standard bedside monitoring system (MP70, Philips Health Care Inc., USA). These data were initially stored at a central station (Philips Health Care Inc.) then transferred to a network drive every 24 hours during the study period. In addition, numerical physiological data such as heart rate were exported continuously from the same MP70 in standard HL7 format at a frequency of 0.2 Hz. These numerical data were stored on the hospital network and then exported manually to the data processing workstation. Regional oxygenation data (CSrO2) were recorded from cerebral sensor, placed on the frontal area, using an INVOS Oximeter (Somanetics Corporation, Troy, MI, USA), and saved as 0.2 Hz continuous data streams. Finally, ventilator data were acquired from the ventilator's RS-232 interface (Servo-i, MAQUET Medical Systems, USA). Data from the INVOS oximeter and ventilator were recorded and stored on a portable personal computer and then transported to the data processing workstation. Simultaneously, continuous EEG monitoring data was recorded for the entire study duration. All recording devices were time-synchronized with the hospital's electronic clock.”
Comment 5: 5) please clarify “Endotracheal (ET) suction events were used as the guide to extract the data needed for this study”
Response: The data used for analysis was originally extracted for our previously published study (Chegondi M, Francis T, Lin WC, Naqvi S, Raszynski A, Totapally BR. Effects of Closed Endotracheal Suctioning on Systemic and Cerebral Oxygenation and Hemodynamics in Children. Pediatr Crit Care Med. 2018;19:e23-e30.). ). For that study the data segments extracted were in relation to endotracheal suctioning as per the objectives of that study.
Comment 6: 6) please explain the reason of 30 minutes testing referring to “Thirty minutes of MAP and CSrO2 were extracted before and after the time point of each ET suctioning event."
Response: We chose 30 minute recording segments of MAP and CsrO2 to increase the ability to track the continuous data and to enable quality data by avoiding motion or cardiorespiratory artifacts, which is a major problems with NIRS monitor.
Comment 7: 7) please explain what does mean “temporal correlation analysis”
Response: Temporal correlation is a time domain analysis of various indices such as MAP and CrSO2 to derive coefficient values over a brief time averages using a longer time data window. For example in our study we used 5-minute averages of MAP and CrSO2 to derive COx value from the 30-minute data window.
Comment 8: 8) please give more technical details referring to "The COX was calculated from each epoch recording of MAP and CrSO2 using the Pearson correlation analysis"
Response: We apologize for not being clear.
To avoid confusion, we deleted the following from the data acquisition and processing section then added to statistical analysis section:
“The COX was calculated from each epoch recording of MAP and CrSO2 using the Pearson correlation analysis, which represented the AR status of the study subject during the recording period. The correlation coefficient derived from Pearson correlation was the concordance level.”
Comment 9: 9) please give technical data related to EEG device and NIRS device. Also, please clarify the problem of synchronization between EEG and NIRS devices.
Response: We have used Natus System for the vEEG machine (Natus Medical Inc. Middleton, WI, USA) and INVOS Oximeter for the NIRS (Somanetics Corporation, Troy, MI, USA). The time clock on vEEG machine and NIRS were adjusted to the hospital clock which all our hospital computers were synchronized. With this synchronization, times on all individual monitors and devices had the same time without any minute discrepancy. We have added these technical details in the methods section under the data acquisition.
Comment 10: 10) please clarify "EEG normal and EEG abnormal" (the criteria for classification)
Response:
We classified normal rhythm and sedation related EEG suppression patterns as normal EEG and burst suppression, periodic lateral epileptiform discharges (PLEDs), and increased activity other than seizures were included in abnormal EEG patterns. We have used the following source for the EEG classification:
Britton JW, Frey LC, Hopp JLet al., authors; St. Louis EK, Frey LC, editors. Electroencephalography (EEG): An Introductory Text and Atlas of Normal and Abnormal Findings in Adults, Children, and Infants [Internet]. Chicago: American Epilepsy Society; 2016. Available from: https://www.ncbi.nlm.nih.gov/books/NBK390354
Comment 11: 11) please explain the reason of application of "Mann-Whitney U test or Median test"
Response: Since we have only 2 groups with non-parametric data, Mann-Whitney U test was used. We have deleted the median test in the manuscript.
Comment 12: 12) please clarify description given in lines 112-115
Response: To derive the concordance ( R) value, we used the 5-minute epoch from the 30 minute of MAP and CrSO2 data window. Then we calculated the average R value for each 30-minute segment and this in both EEG groups.
Comment 13: 13) please clarify relation between “16 sedated and ventilated children” and "Continuous recordings of MAP and CSrO2 by NIRS were extracted from 332 ET suction events" and "From these 332 ET suction events, a total of 580 EEG recordings (30 minutes each) were available for review" and “for 96 hours" (line 15)
Response: We apologize for not being clear.
Here we were trying to explain how we got the total number of EEG recordings suitable for the analysis. From our cohort of 16 sedated and mechanically ventilated children, two 30-minute periods of observation conducted before and after each of 332 ET suctioning events. This yielded 664 30-minute periods of observations. However, we have excluded the observations if there were any missing or incomplete EEG data during the entire 30 minutes of the observation period. As a result, we had 580 EEG recordings for the analysis. Among these 580 EEG recordings, we have excluded missing or incomplete NIRS or ABP variables during the entire 30 minutes of the observation period. At the end, we had 382 EEG recordings with NIRS and ABP data to measure the COx value.
We have included this in the results section.
Comment 14: 14) please clarify description given in line 139
Response: Line 139 is the footnote description for table 2. We changed the font size to highlight this.
Comment 15: 15) please update Introduction by declaring whether method proposed in this study is novel one
Response: The methodology is not novel. This has been used in children with seizures but not for nonictal EEG patterns. We have already mentioned this in the introduction line 56-59 as-
“Multimodal neuromonitoring using EEG and NIRS was used previously in children with seizures to evaluate the neurovascular coupling alterations [15]. However, the association of EEG abnormalities with AR in critically ill children without ictal changes is not known.”
Comment 16: 16) please explain “PaCO2” (line 195)
Response: Partial pressure of carbon dioxide (PCO2) is a potent cerebral vasodilator, and changes in PCO2 levels will ultimately affect cerebral autoregulation.
Comment 17: 17) please specify Conclusions
Response: Thank you for the suggestion
We have added the following sentence to the conclusions-
“The COx values were low in critically ill sedated children without neurological injury or in children with status epilepticus during seizure-free intervals, indicating preserved cerebral autoregulatory function.”
Comment 18: 18) please clarify descriptions given in line 24-25
Response: The analysis showed that the number of 5 minute epochs derived R values from the 30 minute segments data were greater than the cutoff values of 0.2, 0.3 and 0.4 were similar in both EEG groups. Which indicates that the abnormal EEG patterns doesn’t have an affect on cerebral AR compared to normal EEG patterns.
Comment 19: 19) please update description of References
Response: Thank you for the suggestion.
We rectified the issue of the references.
Round 2
Reviewer 1 Report
All questions are answered properly.